# Co-immobilization of whole cells and enzymes by covalent organic framework for biocatalysis process intensification

Dong Zheng[1], Yunlong Zheng[1], Junjie Tan[1], Zhenjie Zhang [2], He Huang [3] & Yao Chen [1,4,5] ✉

Co-immobilization of cells and enzymes is often essential for the cascade biocatalytic processes of industrial-scale feasibility but remains a vast challenge. Herein, we create a facile co-immobilization platform integrating enzymes and cells in covalent organic frameworks (COFs) to realize the highly efficient cascade of inulinase and *E. coli* for bioconversion of natural products. Enzymes can be uniformly immobilized in the COF armor, which coats on the cell surface to produce cascade biocatalysts with high efficiency, stability and recyclability. Furthermore, this one-pot in situ synthesis process facilitates a gram-scale fabrication of enzyme-cell biocatalysts, which can generate a continuous-flow device conversing inulin to D-allulose, achieving space-time yield of 161.28 g L$^{-1}$ d$^{-1}$ and high stability (remaining >90% initial catalytic efficiency after 7 days of continuous reaction). The created platform is applied for various cells (e.g., *E. coli*, Yeast) and enzymes, demonstrating excellent universality. This study paves a pathway to break the bottleneck of extra- and intracellular catalysis, creates a high-performance and customizable platform for enzyme-cell cascade biomanufacturing, and expands the scope of biocatalysis process intensification.

Nowadays, green biomanufacturing is rapidly emerging as a revolutionary production paradigm across multiple domains due to its unparalleled superiorities, including low energy consumption, high enantioselectivity, and environmentally friendly fashions[1–4]. Cell-free enzymes and whole-cells are the 'core chip' of biomanufacturing as highly desirable and versatile biocatalysts because of their regio- and stereoselectivity, high efficiency, and substrate specificity[5–7]. However, great challenges hinder the industrial application of both biocatalysts. Enzymatic catalysis, for instance, may suffer from high cost, low stability, and difficulty in recovery[8]. Whole cell is primarily constrained in low transformation efficiency due to inadequate intracellular mass transport and extracellular transmembrane restriction on large substrates[9,10]. In addition, in numerous instances, enzymes are not

expressible directly in the cells that cooperate functionally with enzymes, necessitating the utilization of extracellular enzymes to achieve cascade reactions with cells[11–13]. Hence, integrating enzymes and cells to leverage their respective strengths is essential and challenging in biomanufacture[14–16]. However, Simply putting them together in solution usually lead to unsatisfactory result attributed to the stability issue and the limitations of mass transfer between enzymes and cells[12,17]. Integrating enzymes and cells on the carrier using immobilization technology can hold on to long-lasting stability and create efficient substrate pathways, allowing them for continuous flow reactions that are more relevant to practical applications[18–20]. Therefore, the immobilization strategy can integrate enzymes and cells to overcome the above challenge and realize biocatalysis process

[1]State Key Laboratory of Medicinal Chemical Biology, College of Pharmacy, Nankai University, Tianjin 300071, China. [2]College of Chemistry, Nankai University, Tianjin 300071, China. [3]School of Food Science and Pharmaceutical Engineering, Nanjing Normal University, 2 Xuelin Road, Nanjing 210023, China. [4]Key Laboratory of Biopharmaceutical Preparation and Delivery, State Key Laboratory of Biochemical Engineering, Chinese Academy of Sciences, Beijing 100190, China. [5]Haihe Laboratory of Synthetic Biology, Tianjin 300308, China. ✉e-mail: chenyao@nankai.edu.cn

intensification. However, the study in this field is still in its infancy, and it is urgently demanded to develop strategies and carriers to achieve the efficient co-immobilization of enzymes and cells.

The immobilization of cells and enzymes separately has currently been extensively studied[21–24], while the co-immobilization of enzymes and cells still confronts great challenges. Biocatalysis usually conducts in an aqueous solution with ions, which may result in the unintended dissociation of co-immobilized carriers that rely on weak interactions (e.g., calcium alginate or metal-organic frameworks, MOFs), leading to catalyst leakage and compromising system stability. Moreover, carrier disintegration can introduce impurities, especially metal ions, leading to difficulty in separation and disruption/contamination of final products. In addition, current carriers are usually non-/micro-porous, hindering mass transfer and catalytic efficiency[12,25]. Recently, covalent organic frameworks (COFs) have emerged as a new generation of porous carriers due to their distinct characteristics, such as well-defined and designable structures, high porosity, tunable pore aperture, and facile functionality[26,27]. However, the synthesis conditions of COFs are dominated by the solvothermal method, which requires harsh conditions for biocatalysts (high temperature and organic solvents). We recently created an in situ assembly method for enzyme immobilization by COFs at the aqueous solution and room temperature, demonstrating high stability, efficiency and versatility[28,29].

To achieve the integration of various catalytic components (such as cells and enzymes) to enhance biocatalytic processes, in this work, we integrate inulinase (INU) and *E. coli* cells by a COF immobilization platform to obtain enzyme-cell co-immobilization biocatalysts (enzyme&cell@COFs) (Fig. 1), contribute to stable and efficient catalysis, and achieve continuous-flow reaction. We believe this co-immobilization platform is broadly applicable and may serve as an immobilization model for enzyme and whole-cell biocatalysts in the future.

## Results and discussion
### Rational design and preparation of COF platform for immobilization

Our previous study revealed that the acylhydrazone-linked COFs (e.g., NKCOF-98) with a skeleton of COF-42 could serve as excellent carriers for enzyme immobilization due to its mild synthesis conditions (room temperature and aqueous solution) and co-reaction of aldehyde groups of COF monomers with amino groups in enzymes[29]. Therefore, COF-42 analogs can be good candidates for immobilizing other biosystems, such as cells. Due to the complex components of the whole-cell walls (e.g., -COOH, -NH₂, and aromatic structure, lipopeptides, fatty acids, or membrane proteins)[30–32], the cell walls usually exhibit amphiphilic features. Thus, it is anticipated that the carrier materials possessing both hydrophilic and hydrophobic constituents will facilitate a more enhanced integration with whole cells. Here, we designed an amphiphilic monomer, 2-(but-3-en-1-yloxy)−5-(2-methoxyethoxy)-terephthalohydrazide (BYTH) with both hydrophilic alkoxy chain and hydrophobic alkyl chain (Supplementary Method 1, Supplementary Fig. 1). Reaction of BYTH and 1,3,5-triformylbenzene (TB) using acetic acid as a catalyst can create a COF, NKCOF-141, which possessed the skeleton of COF-42 proved by Powder X-Ray Diffraction (PXRD) pattern (Supplementary Fig. 2). The formation of acylhydrazone linkage was verified by Solid State Nuclear Magnetic Resonance and Fourier Transform Infrared (FT-IR) spectrum (Supplementary Figs. 3 and 4). D-allulose, being a rare sugar, holds promising prospects in the food, healthcare, and pharmaceutical industries[33–36]. Due to limitations in purification of D-allulose 3-epimerase (DAE), and catalytic conditions of DAE and inulinase (INU) from inulin to D-allulose, the cooperation of INU and whole *E. coli* cells expressing DAE (E) may be a solution. Since the *E. coli* cells expressing DAE (E) are more stable in phosphate buffer saline (Supplementary Method 2), we used phosphate buffer saline (PBS) as the reaction medium. *E. coli* cells were added in one pot with TB and BYTH, producing a cell-encapsulating COF composite (E@NKCOF-141, Fig. 2a), which retained the crystalline structures of NKCOF-141 revealed by PXRD data (Fig. 2b). Moreover, the analysis of N₂ sorption isotherms at 77 K demonstrated a decreased BET surface area from 275 m² g⁻¹ to 154 m² g⁻¹ owing to the encapsulated whole cells, which increased the material density (Fig. 2c). The density functional theory (DFT) pore size distribution analysis evidenced that E@NKCOF-141 exhibited similar pore sizes (~1.8 nm) as pristine NKCOF-141 (Fig. 2c). These results demonstrated that the whole cells did not block the pores of COF carriers, which provided the basis for small molecule substrates to enter the cell. After in situ immobilization of cells by NKCOF-141 (Fig. 2d), the amide I peak, which primarily corresponded to the C=O stretching vibration, experienced a blue shift from 1633 cm⁻¹ (free E) to 1660 cm⁻¹ (E@NKCOF-141), implying a strong interaction of NKCOF-141 on the cell surface. To further evaluate whether NKCOF-141 could uniformly coat on the surface of cells, the free cells and E@NKCOF-141 were compared using transmission electron microscopy (TEM). As shown in Supplementary Fig. 5, the surface of the cells became roughened after NKCOF-141 immobilization, and COF particles were uniformly wrapped around the whole cell. The coating of NKCOF-141 was also clearly observed in the TEM

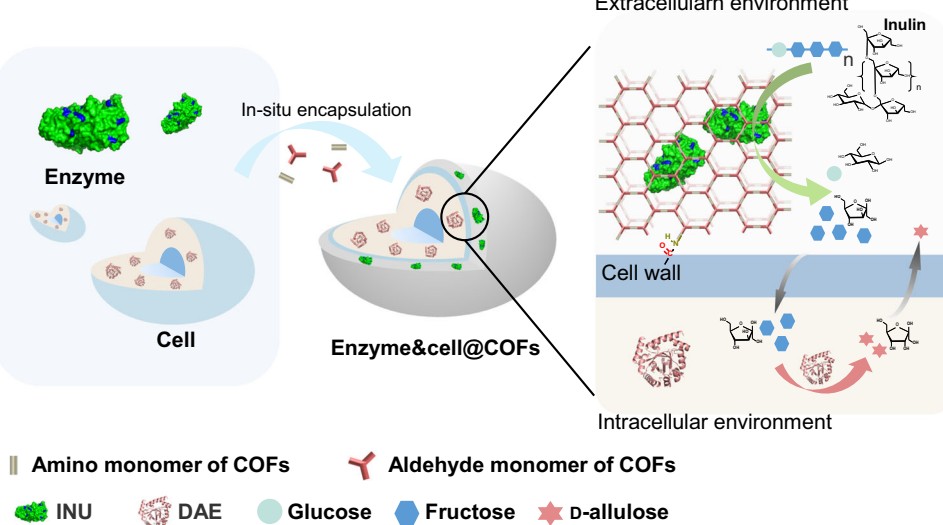

**Fig. 1 | Construction of enzyme&cell@COFs co-immobilization biocatalysts.** Schematic diagram of the in situ assembly approach of enzyme&cell@COFs.

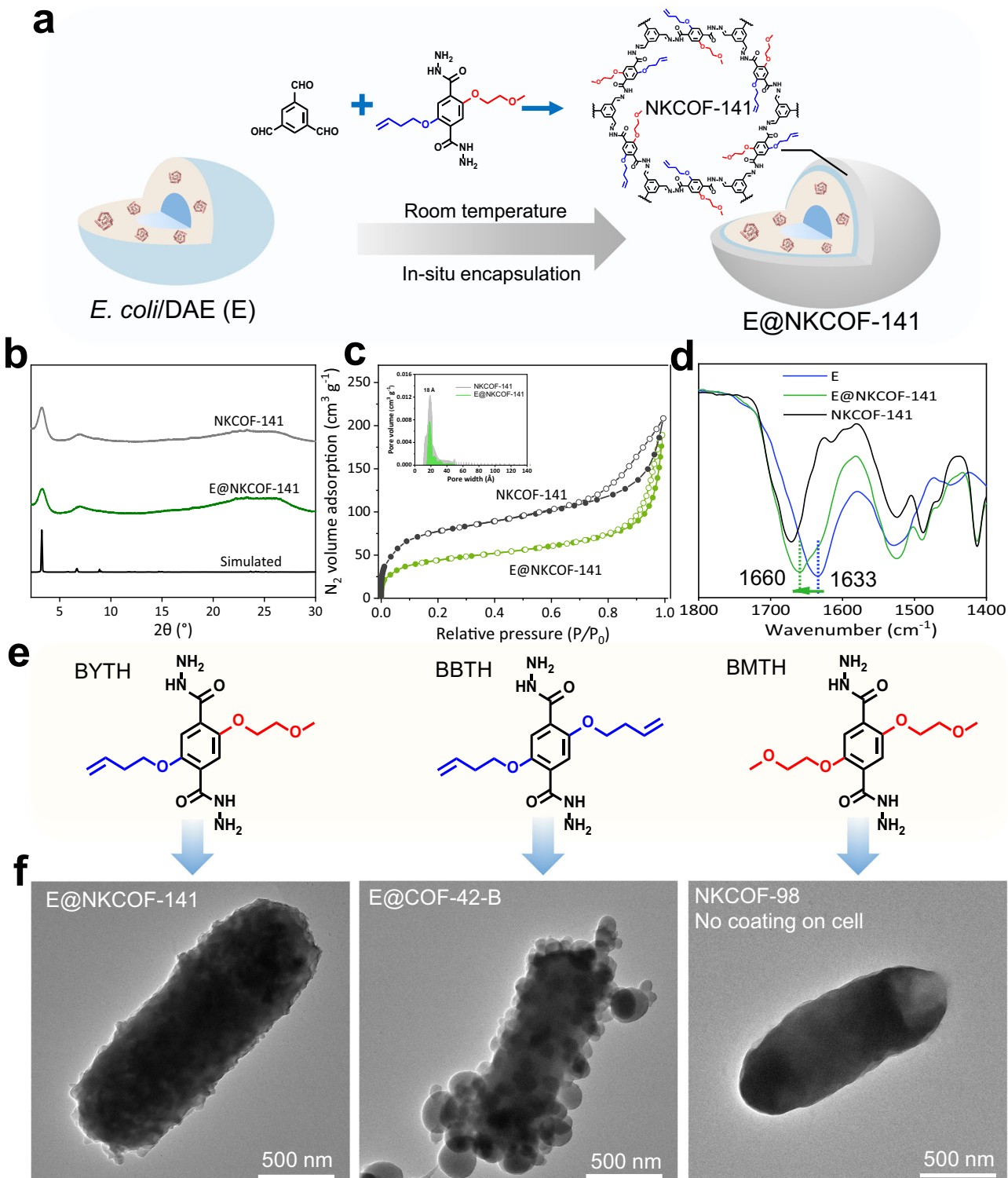

**Fig. 2 | Synthesis and characterization of E@COFs. a** Illustration of the preparation for E@NKCOF-141. **b** PXRD patterns of NKCOF-141, E@NKCOF-141, and simulated NKCOF-141. **c** N$_2$ sorption isotherms of NKCOF-141 and E@NKCOF-141, and the corresponding pore size distribution. **d** FT-IR spectra (vector normalized in the spectra region 1400–1800 cm$^{-1}$) of NKCOF-141, E, and E@NKCOF-141. **e** Amino monomers of COFs. **f** TEM images of E@NKCOF-141, E@COF-42-B and product of NKCOF-98 without coating on cells. Each experiment was repeated independently three times with similar results ($n = 3$).

micrograph of the microtomed E@NKCOF-141, and the average thickness was around 20 nm (Supplementary Fig. 6). The activity analysis of E@NKCOF-141 revealed that the adding amount of acetic acid as a catalyst would affect cell activity (Supplementary Fig. 7). After optimization, it was found that the addition of 10.5 mM of acetic acid

afforded a high relative activity (97%) while ensuring the crystallinity of E@NKCOF-141 (Supplementary Figs. 7 and 8). The biocompatibility was assessed through live/dead assays in the optimized immobilization system[23]. From Supplementary Fig. 9a, b, the ratio of living cells (green) and dead cells (red) among E@NKCOF-141 was similar to the

free E. Although E@NKCOF-141 showed a delayed growth curve compared to free *E. coli* cells, E@NKCOF-141 exhibited consistent growth viability with free E (Supplementary Fig. 9c). These findings highlighted that the immobilization of E by NKCOF-141 was mild enough to allow cell division even after the coating procedure. In addition, to highlight the advantage of amphiphilic COFs for cell immobilization, we synthesized two analogs of NKCOF-141, NKCOF-98 (reaction of 2,5-bis (2-methoxyethoxy) terephthalohydrazide (BMTH, Supplementary Fig. 10) and TB) and COF-42-B (reaction of 2,5-bis(but-3-en-1-yloxy) terephthalohydrazide (BBTH, Supplementary Fig. 11) and TB) as comparisons (Fig. 2e, Supplementary Method 3 and 4). Both COFs could form crystalline and porous phases after adding cells during the material formation process (Supplementary Figs. 12–14). However, TEM images in Fig. 2f showed that NKCOF-98 did not coat the cells, whereas COF-42-B could form a coating on the cells, but the uniformity was worse than that of NKCOF-141. Considering the crystallinity and morphology of E@NKCOFs, we chose the E@NKCOF-141 system with good crystallinity and uniform coating for the following studies.

Subsequently, we immobilized Inulinase (INU) with the same synthesis condition as the cell immobilization to produce the composite material (named INU@NKCOF-141, Fig. 3a). The PXRD, FT-IR, $N_2$ sorption and pore size distribution results confirmed the successful fabrication of INU@NKCOF-141 (Supplementary Figs. 15–17). To investigate the distribution of INU, fluorescein isothiocyanate (FITC)-labeled INU (FITC-INU) was immobilized and tested using confocal laser scanning microscopy (CLSM). The size of INU was ~ 3.8 nm × 4.7 nm × 7.3 nm, which was much larger than that of NKCOF-141 (~1.8 nm), so INU would not be directly adsorbed by NKCOF-141 (Supplementary Fig. 18). The CLSM images of FITC-INU@NKCOF-141 indicated that FITC-INU (green) was uniformly dispersed throughout the formed composites (Fig. 3b). INU@NKCOF-141 retained the majority of enzymatic activity (>80%) with a relatively low loading of INU (0.155 g g$^{-1}$, Supplementary Fig. 19). The enzyme loading amount may affect the subsequent cascade reaction, so it was essential to increase the INU loading and regulate the catalytic effect. Our previous research revealed that the amino groups in

the enzyme could react with the aldehyde monomer to promote better integration during the in situ immobilization process[29]. The relatively low loading of INU might be attributed to the lack of terminal amino groups (-$NH_2$) of enzymes (Supplementary Table 1). Thereby, we pre-incubated INU with EDC/S-NHS to activate the -COOH of enzymes, and then reacted with -$NH_2$ groups of BYTH to form BYTH-modified INU (INU-$NH_2$) (retaining 87% of its initial enzyme activity, Supplementary Figs. 20 and 21) and obtained INU-$NH_2$@NKCOF-141 with highly-loading and activity (Supplementary Fig. 22). The INU loading could be significantly increased from 0.155 g g$^{-1}$ to 2.291 g g$^{-1}$ (Fig. 3c). The increase of enzyme loading capacity was also confirmed by CLSM. INU-$NH_2$@NKCOF-141 had an enhanced fluorescent signal than INU@NKCOF-141 under identical imaging operations, indicating higher protein encapsulation efficiency[37]. Overall, NKCOF-141 demonstrated excellent encapsulation efficiency for INU-$NH_2$ and cells with decent activity (84% and 97%, separately), which undoubtedly provided a basis foundation for the construction of enzyme-cell co-immobilization biocatalysts.

## Construction of enzyme-cell co-immobilization biocatalysts

We then conducted the co-immobilization of *E. coli*/DAE (E) and FITC-INU-$NH_2$ via NKCOF-141 by one-pot synthesis. During the process, we found the competitive immobilization between E (red) and FITC-INU-$NH_2$ (green) (Supplementary Fig. 23), and NKCOF-141 preferred to immobilize INU-$NH_2$ possibly owing to the pre-assembly with BYTH (Supplementary Fig. 24). Therefore, we also pre-treated the cells with BYTH under EDC/S-NHS, and the treated cells (E-$NH_2$) exhibited consistent growth viability and enzyme activity with regular *E. coli* counterparts (Supplementary Fig. 25). When E-$NH_2$ was co-immobilized with FITC-INU-$NH_2$ using NKCOF-141, it was seen that FITC-INU-$NH_2$ arranged more compactly on E-$NH_2$ surface (Supplementary Fig. 26). And, in the enzyme-cell incubation system in the absence of COF, the INU-$NH_2$ did not bind on the cell surface to form enzyme-bacteria assemblies (Supplementary Fig. 27). As shown in Fig. 4a, the CLSM images obviously showed that FITC-INU-$NH_2$ and E-$NH_2$ were co-immobilized by NKCOF-141, and the cross-sections (Z-axis) of 3D view

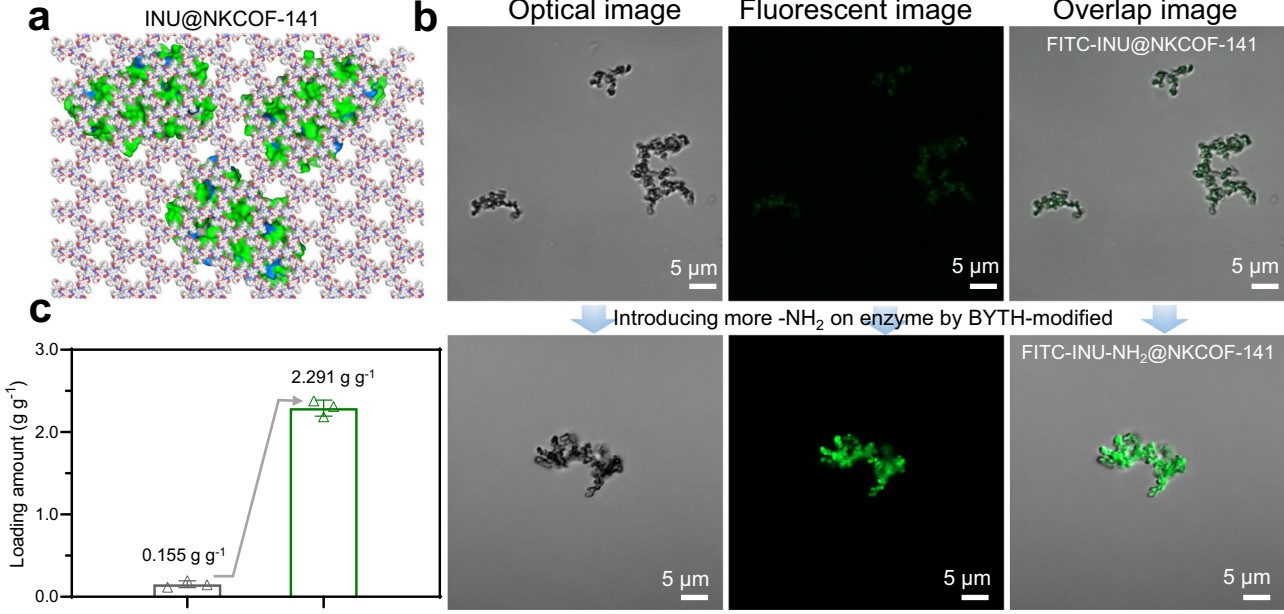

**Fig. 3 | Synthesis and characterizations of INU@NKCOF-141. a** Structure illustration for INU@NKCOF-141. **b** CLSM images of FITC-INU@NKCOF-141 and FITC-INU-$NH_2$@NKCOF-141, excitation at 488 nm and tracking the fluorescence at 520 nm. Each experiment was repeated independently three times with similar results (*n* = 3). **c** The loading amounts of INU@NKCOF-141 and INU-$NH_2$@NKCOF-141. Error bars mean ± s.d. received from three independent experiments (*n* = 3). Source data are provided as a Source Data file.

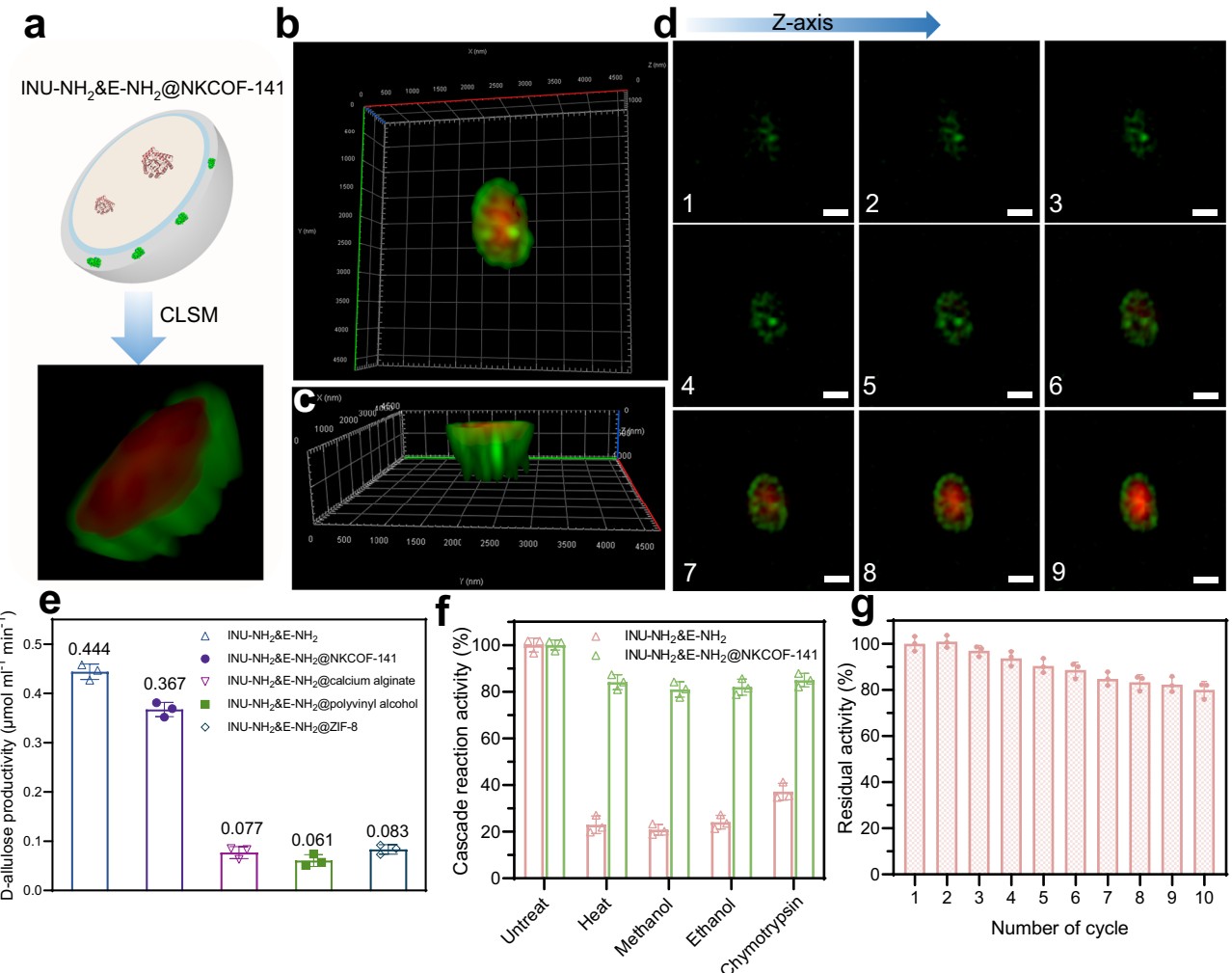

**Fig. 4 | Co-immobilization of enzyme and cell by NKCOF-141 and characterizations. a** Schematic illustration and reconstructed 3D CLSM image of FITC-INU-NH$_2$&E-NH$_2$@NKCOF-141. The 3D top view (**b**) and the 3D side view (**c**) CLSM images of FITC-INU-NH$_2$&E-NH$_2$@NKCOF-141. **d** CLSM images of the FITC-INU-NH$_2$&E-NH$_2$@NKCOF-141 at a series of focal planes measured every 0.2 μm along the Z-axis. The cells were labeled with propidium iodide (red). Scale bar: 0.5 μm. Each experiment was repeated independently three times with similar results (*n* = 3). **e** The catalytic activity of immobilized INU-NH$_2$&E-NH$_2$ in different materials, and co-immobilized systems were loaded with ~ 4 mg of cells and 1 mg of enzyme. **f** Stability of INU-NH$_2$&E-NH$_2$ and INU-NH$_2$&E-NH$_2$@NKCOF-141 after treatment in heat (60 °C) for 20 min, methanol for 40 min and ethanol for 30 min, and 15 mg mL$^{-1}$ chymotrypsin for 30 min. **g** Recyclability of INU-NH$_2$&E-NH$_2$@NKCOF−141, in which the initial content of D-allulose was set as 100%. All error bars mean ± s.d. received from three independent experiments (*n* = 3). Source data are provided as a Source Data file.

CLSM images showed that FITC-INU-NH$_2$ was uniformly immobilized on the cell surface by NKCOF-141 (Fig. 4b–d). The PXRD, N$_2$ sorption, FT-IR and TEM results further verified the successful preparation of INU-NH$_2$&E-NH$_2$@NKCOF-141 (Supplementary Figs. 28–31). The cascade reaction was evaluated by a typical method from inulin to D-allulose, and we adjusted the ratio of INU-NH$_2$ and E-NH$_2$, cascade reaction temperature and pH to determine the reaction conditions by D-allulose productivity in 10 min (Supplementary Method 5). The optimal ratio of INU-NH$_2$ and E-NH$_2$ was 1:4 in the co-immobilization system (Supplementary Fig. 32), and 50 °C, pH 6.5 were found to be the optimal conditions (Supplementary Fig. 33). The results showed that under optimal conditions, INU-NH$_2$&E-NH$_2$@NKCOF-141 possessed high enzyme loading (1.34 g g$^{-1}$) and demonstrated 0.367 μmol mL$^{-1}$ min$^{-1}$ for the production of D-allulose, which was close to the production efficiency of free INU-NH$_2$&E-NH$_2$ system (Supplementary Fig. 34). For comparison, the immobilization carriers, such as calcium alginate, polyvinyl alcohol, and ZIF-8 were also evaluated by integrating INU-NH$_2$ and E-NH$_2$ (Supplementary Method 6–8), leading to production efficiency of D-allulose

< 0.10 μmol mL$^{-1}$ min$^{-1}$, which was much lower than INU-NH$_2$&E-NH$_2$@NKCOF-141 under the same enzyme-cell loading amount (Fig. 4e). The excellent cascade activity of INU-NH$_2$&E-NH$_2$@NKCOF-141 was possibly attributed to the following reasons: (i) compact binding of enzymes and cells to optimize substrate pathways, (ii) uniform pores to benefit the diffusion, (iii) avoiding the toxicity of metals to cells and enzymes[25,38]. These findings provide compelling evidence supporting the good biocompatibility and activity retention capabilities of NKCOF-141 as an immobilization carrier for enzymes and cells.

INU-NH$_2$&E-NH$_2$@NKCOF-141 could enhance biocatalysts' industrial properties, such as half-life ($T_{1/2}$), stability and reusability. To be specific, we systematically investigated the stability and recyclability of INU-NH$_2$&E-NH$_2$@NKCOF-141. We firstly assessed the $T_{1/2}$ at 50 °C, and the ~$T_{1/2}$ value was 990 min for INU-NH$_2$&E-NH$_2$@NKCOF-141, which was 7.56-fold higher than that of the free INU-NH$_2$&E-NH$_2$ (Supplementary Table 2, Supplementary Fig. 35). We further analyzed the resistances of INU-NH$_2$&E-NH$_2$@NKCOF-141 against perturbation environments such as high temperatures, organic solvents, and

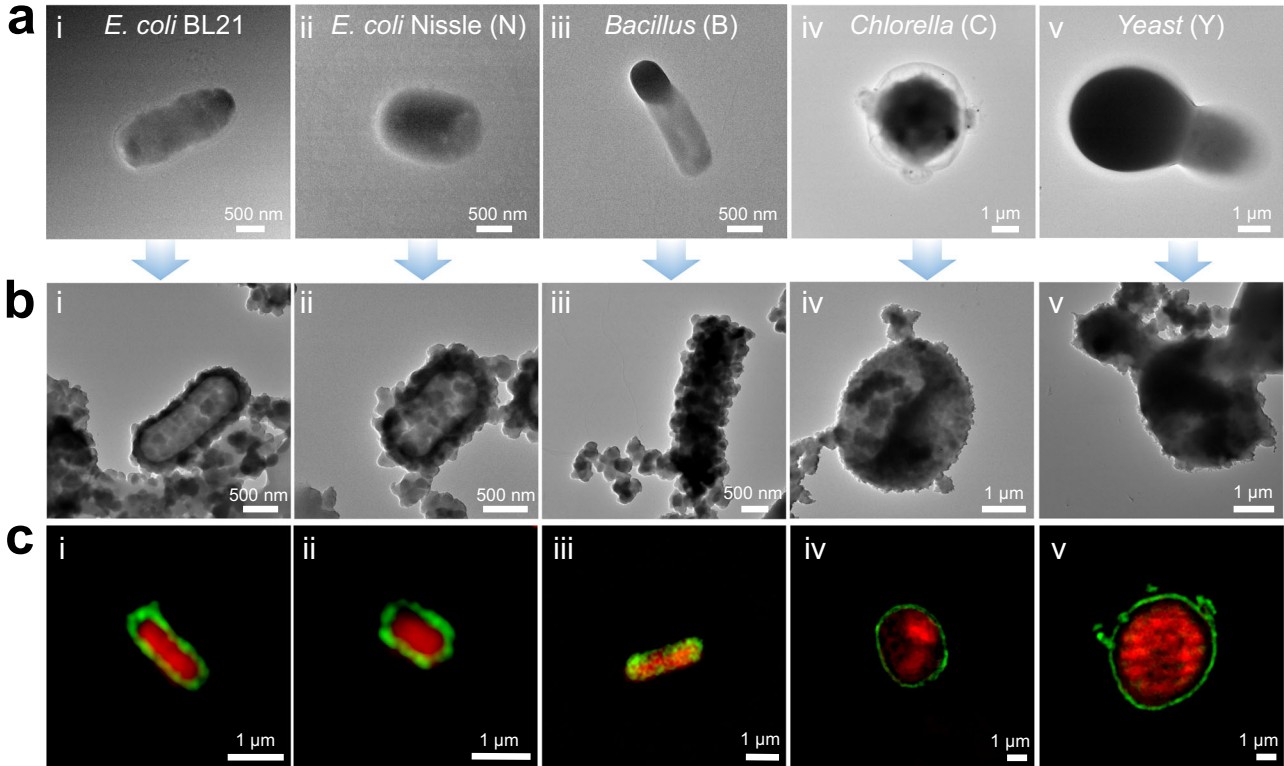

**Fig. 5 | The generality of enzyme-cell co-immobilization by NKCOF-141. a** TEM images of whole cells of various types, (i) *E. coli* BL21/lipase, (ii) *E. coli* Nissle 1917, (iii) *Bacillus subtilis*, (iv) Chlorella, (v) Yeast. **b** TEM images of BSA&cells-NH$_2$@NKCOF-141 through co-immobilization. **c** The CLSM images of (i) FITC-BSA&E/lipase-NH$_2$@NKCOF-141, (ii) FITC-BSA&N-NH$_2$@NKCOF-141, (iii) FITC-BSA&B-NH$_2$@NKCOF-141, (iv) FITC-BSA&C-NH$_2$@NKCOF-141 and (v) FITC-BSA&Y-NH$_2$@NKCOF-141. The cells were labeled with propidium iodide (red). Each experiment was repeated independently three times with similar results (*n* = 3).

proteases (Fig. 4f). After treatment with 60 °C for 20 min, INU-NH$_2$&E-NH$_2$@NKCOF-141 was seen to retain a cascade activity of 84%. In contrast, the cascade activity of free INU-NH$_2$&E-NH$_2$ retained only 23%. After treatment with methanol for 40 min and ethanol for 30 min, the cascade activity of INU-NH$_2$&E-NH$_2$@NKCOF-141 retained >80%, while free INU-NH$_2$&E-NH$_2$ was significantly reduced (below 25%). A similar result was observed after treatment of Chymotrypsin trypsin (15 mg mL$^{-1}$) for 30 min: INU-NH$_2$&E-NH$_2$@NKCOF-141 retained 85% of cascade activity, while free INU-NH$_2$&E-NH$_2$ decreased >60% of cascade activity. In addition, the protective efficiencies were enhanced by the increase in the integrality and thickness of the shell, which was consistent with previous reports (Supplementary Fig. 36)[39–41]. These results revealed that the co-immobilization of NKCOF-141 effectively improved the tolerance against external disturbance. As shown in Fig. 4g, INU-NH$_2$&E-NH$_2$@NKCOF-141 exhibited high recyclability, and maintained >80% of cascade activity after 10 cycles. All these results highlighted that co-immobilization by COFs to integrate whole cells and enzymes had superior activity and stability, providing great potential for the practical applications of these biocatalysts.

To verify the generality of our co-immobilization platform, we expanded the cells from bacteria to fungi, such as Chlorella and Yeast, and used other proteins (BSA). We used *Escherichia coli* BL21/lipase (E/lipase) (Supplementary Method 9), *Escherichia coli* Nissle (N), *Bacillus subtilis* (B), *Chlorella pyrenoidosa* (C) and *Saccharomyces cerevisiae* (Yeast, Y) as the immobilized whole cells. Subsequently, cell-NH$_2$ and BSA were co-immobilized via NKCOF-141 (Supplementary Figs. 37 and 38). As revealed by PXRD patterns (Supplementary Fig. 39), all co-immobilization composites possessed crystallinity as good as the pristine NKCOF-141. As shown in Fig. 5a, b, TEM images showed the encapsulation of different types of whole cells by NKCOF-141. CLSM images further illustrated that the immobilized FITC-BSA (green) was tightly packed around the periphery of the whole cell (red) and wrapped around the whole cell (Fig. 5c). It has been demonstrated that FITC-BSA was uniformly immobilized on the surface of *Escherichia coli* expressing different proteins (*E. coli*/DAE and *E. coli*/lipase) by NKCOF-141. In addition, the different strains *Escherichia coli* (*E. coli* BL21 and *E. coli* Nissle) also achieved excellent integration to obtain enzyme-cell co-immobilization, and similar results were obtained for different types of whole cells (*E. coli*, *Bacillus subtilis*, *Chlorella pyrenoidosa*, Yeast). These findings demonstrated that NKCOF-141 is indeed a versatile platform to enable simultaneous co-immobilization of the cells and enzymes.

## The continuous-flow reaction for production of D-allulose

Due to the enhanced stability and processability of biological components offered by co-immobilization, along with the expanded range of operating conditions, we successfully constructed a device and assessed its catalytic performance in a continuous-flow reaction, which is regarded as a more advantageous approach for industrial-scale production[42]. INU-NH$_2$&E-NH$_2$@NKCOF-141 composite catalysts were packed into the column, and the inulin solution was pumped into the inlet of the system at a flow rate of 0.1 mL min$^{-1}$ (Fig. 6a, Supplementary Fig. 40). The outflow, which contained the reaction products, was collected continuously for HPLC analysis. By optimizing the flow rate (Supplementary Fig. 41), at a flow rate of 0.2 mL min$^{-1}$, the yield of D-allulose was 22.4%, the maximum yield achievable at the reaction equilibrium. When the flow rate was further increased, the yield of D-allulose showed a decreasing trend, which could be attributed to the shortened contact time and reduced collision efficiency between the reactants and INU-NH$_2$&E-NH$_2$@NKCOF-141 at higher reactant flow rates[43]. Under conditions of 50 °C and a flow rate of 0.2 mL min$^{-1}$, the space-time yield (STY) of D-allulose was up to 161.28 g L$^{-1}$ day$^{-1}$, which

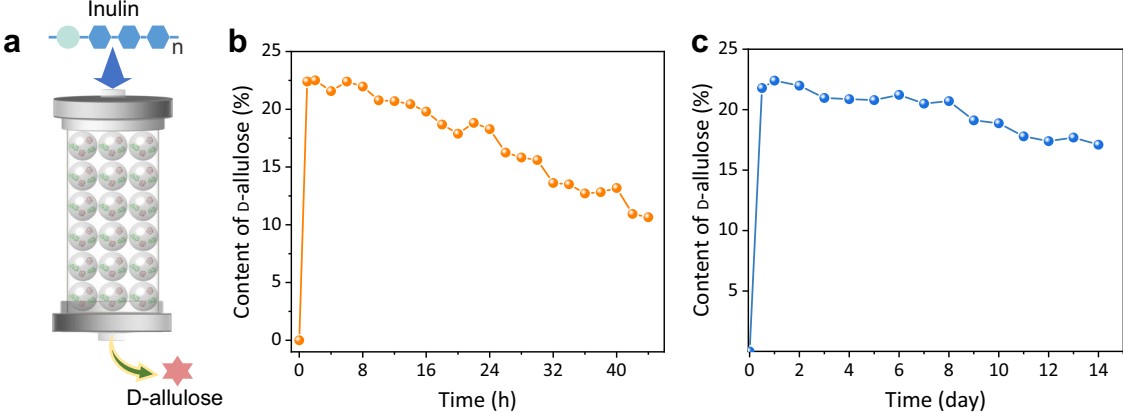

**Fig. 6 | Production of D-allulose by INU-NH₂&E-NH₂@NKCOF-141. a** Schematic diagram of the continuous-flow reaction for the production of D-allulose from inulin by INU-NH₂&E-NH₂@NKCOF-141. **b** Time-dependent content for D-allulose at 50 °C and 0.2 mL min⁻¹. **c** Time-dependent content for D-allulose based continuous-flow reaction of room temperature (30 °C) and 0.1 mL min⁻¹.

was 2.7 times higher than the current highest record (59.4 g L⁻¹ day⁻¹) (Supplementary Table. 3). In parallel, the introduction of NKCOF-141 showcased good stability within the biocatalytic system and maintained an enduring high catalytic efficiency (>20% D-allulose content) over an extended period (0–14 h) (Fig. 6b).

In fact, in industrial applications, heating poses more challenges for equipment and energy consumption. Therefore, we worked on regulating continuous catalytic reactions in device columns at room temperature. Reducing the reaction temperature would lead to a decrease in the enzymatic activity of INU and *E. coli*/DAE, and we balanced the catalytic reaction by adjusting the reaction flow rate to optimize the production of D-allulose. As shown in Supplementary Fig. 42, the catalytic reaction of D-allulose could also reach equilibrium at a lower flow rate (0.1 mL min⁻¹) compared to the 50 °C catalyzed reaction. Although the STY of D-allulose (0.1 mL min⁻¹ up – 80.64 g L⁻¹ day⁻¹) at room temperature was reduced by half compared to that at 50 °C, the reaction exhibited better stability, which was 14 times higher than at 50 °C, and was more energy-efficient and environmentally friendly. Notably, even after 14 days of uninterrupted flow reaction, a space-time yield of D-allulose retained 61.56 g L⁻¹ day⁻¹ (76.3% of the initial STY) (Fig. 6c). These results highlighted that this co-immobilization system realized efficient and stable conversion under room temperature conditions. Moreover, it was notable that the combination of free INU-NH₂ and E-NH₂ did not support a continuous flow reaction, and the combination of INU-NH₂@NKCOF-141 and E-NH₂ showed a significant decrease in the yield of D-allulose at 6 days. In addition, it was found that the content of D-allulose was low, just 16.1%, in combination with INU-NH₂@NKCOF-141 and E-NH₂@NKCOF-141, and INU-NH₂&E-NH₂@NKCOF-141 had a higher production capacity of D-allulose compared to other unitized equipment (Supplementary Fig. 43). All the above results further emphasized the excellent activity and brilliants stability of the NKCOF-141 co-immobilized cell-enzyme in a continuous reaction apparatus.

In summary, we created an enzyme-cell co-immobilization platform for scalable fabrication of highly efficient biocatalysts via facile in situ assembly of cells and enzymes. We realized the integration of inulinase (INU) and *E. coli* cells expressing DAE (E) by NKCOF-141 under mild one-pot synthesis conditions (room temperature and aqueous solution). Due to the integration and porosity of COF materials allowing for rapid diffusion of substrates and products, the biocatalyst of co-immobilization enzyme and cell (INU-NH₂&E-NH₂@NKCOF-141) presented high catalytic efficiency, compared to other co-immobilized systems (such as calcium alginate, polyvinyl alcohol, and ZIF-8). Additionally, INU-NH₂&E-NH₂@NKCOF-141 exhibited high stability, half-life has increased by ~8 times, and excellent reusability (cascade

activity after 10 cycles >80%) attributed to stabilized conformation of enzyme and cellular structure by protective carrier NKCOF-141. Moreover, we also conducted the scale-up synthesis of INU-NH₂&E-NH₂@NKCOF-141 biocatalysts and constructed a continuous reaction apparatus using these biocatalysts. The results demonstrated the productivity of D-allulose could reach up to 161.28 g L⁻¹ day⁻¹, which is the highest record for D-allulose from inulin so far, and showed significant stability (remaining >76% initial catalytic efficiency after 2 weeks of continuous reaction). Furthermore, this enzyme-cell co-immobilization platform demonstrated excellent universality, which could be extended from bacteria to fungi (e.g., *E. coli*, Nissle, *Bacillus*, *Chlorella*, and Yeast) and various proteins (e.g., inulinase, BSA). This study innovates a high-performance and customizable platform to overcome one of the key challenges of biomanufacturing, advances the commercialization of enzyme-cell cascade production of high-value products, and paves an avenue for the process intensification of biocatalysis.

## Methods
### Chemicals
Unless specifically stated otherwise, all materials utilized in the experiments, including solvents, were procured from commercial suppliers and utilized without any additional purification. Fructose, glucose, inulin, D-allulose, and inulinase was purchased from Yuanye (Shanghai, China). FITC-BSA was obtained from Solarbio (Beijing, China). 1-(3-Dimethylaminopropyl)−3-ethylcarbodiimide hydrochloride (EDC) and N-Hydroxysulfosuccinimide sodium salt (S-NHS) were obtained from Bidepharm (Shanghai, China). Fluorescein isothiocyanate (FITC) was obtained from Heowns Biochem Technologies LLC (Tianjin, China).

### Characterization
1H nuclear magnetic resonance (NMR) data were acquired using a Bruker 400 MHz NMR spectrometer and solid state ¹³C NMR was entered on Infinityplus 300 NMR spectrometer. Powder X-ray diffraction (PXRD) measurements were conducted on a D/Max-2500 X-ray diffractometer by applying the powder onto a glass substrate. The scans were performed over 2θ from of 1.5° to 30° with a step size of 0.02°. The low-pressure nitrogen adsorption and desorption isotherms were obtained at 77 K using a Micromeritics ASAP 2460 instrument. Fourier transform infrared spectrophotometer (FT-IR) spectra were collected using a Nicolet IS 10 spectrometer. Transmission electron microscope (TEM) images were recorded using the Talos L120C G2 operating at an acceleration voltage of 120 kV. Confocal Laser Scanning Microscopy (CLSM) images were performed using a

Zeiss LSM 800 confocal microscope equipped with a 63 × objective oil lens, and high-definition confocal images were carried out using a Zeiss Elyra 7 Super-Resolution Imaging System SIM & SMLM. The catalytic process was monitored using a Hitachi UH-5300 UV-vis spectrophotometer. The SpectraMax M2e microplate reader was utilized to measure the absorbance value. Identification of sugars were analyzed by high-performance liquid chromatography (HPLC). Agilent 1260 instrument (USA) with a refractive index detector using a Inertsil NH$_2$ (5 μm, 4.6 × 250 mm, GL Sciences, Japan).

### Synthesis of NKCOF-141

In a typical synthesis, 2-(but-3-en-1-yloxy)−5-(2-methoxyethoxy)-terephthalohydrazide (BYTH, 10 mg) and 1,3,5-triformylbenzene (TB, 3.3 mg) were dissolved in 20 mL aqueous solution. Then, 20 μL acetic acid (17.5 mM) was added into the solution. After reacting at room temperature for 2 h, the precipitate, NKCOF-141, was obtained through centrifugation and then water washing.

### Synthesis of *E. coli*@NKCOF-141

Monomers with TB (3.3 mg) were dissolved in 10 mM PBS (15 mL, pH 7.4). In parallel, 40 mg wet-weight *E. coli*/DAE (E, 2.04 × 10$^5$ CFU/mg) was resuspended in the solution above. 20 μL acetic acid (17.5 mM) was added to the solution to react for 20−30 min. Then, 5 mL of PBS containing BYTH (10.0 mg) was added to the reaction solution. After reacting at room temperature for 2 h, the E@NKCOF-141 was centrifuged and washed with PBS for 2 times.

### Enzymatic activity of DAE

The catalytic activity of *E. coli*/DAE (E) and immobilized E was assayed by measuring the production of D-allulose from D-fructose. The reaction mixtures in PB (50 mM, pH 6.5) contained 10 g L$^{-1}$ D-fructose and 4 mg E (wet weight, 2.04 × 10$^5$ CFU/mg) or E@NKCOF-141 (with the same number of cells). The cell wet weight was measured by weighing a cell pellet that was gained by centrifugation at 6000 × g for 10 min. The reaction was conducted at 50 °C for 10 min in a final volume of 2 mL. And then, the cells were precipitated by centrifugation, and the reaction supernatants were stopped by boiling for 5 min. The D-allulose concentration was assayed using high-performance liquid chromatography (HPLC). One D-allulose production unit (APU) was expressed as the amount of DAE cells that catalyzed the formation of 1 μmol D-allulose from D-fructose per min at pH 6.5 and 50 °C. Each measurement was performed in triplicate, and the results were represented as the means ± standard deviation (SD).

The qualitative and quantitative analysis of D-allulose were determined using high-performance liquid chromatography (HPLC). Agilent 1260 instrument (USA) with a refractive index detector using a Inertsil NH$_2$ (5 μm, 4.6 × 250 mm, GL Sciences, Japan) was taken, acetonitrile (75%) was used as the mobile phase at a flow rate of 1.0 mL min$^{-1}$, and the column was kept at 35 °C. Different concentrations of D-allulose standard (1, 2, 3, 4, and 5 mg mL$^{-1}$) were utilized to plot the standard curve. All measurements were performed in triplicate.

### Synthesis of INU@NKCOF-141

The typical method consisted of introducing 30 mg weighed enzyme into the 20 mL aqueous solution, which included 10 mg BYTH and 3.3 mg of TB, and subsequently adding acetic acid (20 μL, 10.5 mM). After 2 h at room temperature, the INU@NKCOF-141 was collected by centrifugation. Subsequently, the obtained product was washed with PBS for 3 times prior to the evaluation of its activity. Following the encapsulation and washing procedure, the supernatant from each step was gathered to quantify the protein concentration using the standard Bradford assay protocol. Prior to this, calibration curves were created using BSA. The loading efficiency (LE%) and loading capacity (LC, g g$^{-1}$)

were determined by the subsequent conclusion.

$$LE\% = (\text{total protein} - \text{protein in supernatant})/\text{total protein} \times 100\% \quad (1)$$

$$LC = \text{protein loaded into materials}/\text{weight of materials} \quad (2)$$

### Enzymatic activity of INU

INU or INU@NKCOF-141 activity was determined in a standard reaction that contained 2 mL phosphate buffer (PB) of 10 g L$^{-1}$ inulin (Yuanye, Shanghai, China) and 1 mg INU or INU@NKCOF-141 (containing the same amount of enzyme) at 50 °C for 10 min. The DNS method was used to assay the reduced sugar yield. In the DNS method, 200 μL of diluted reaction solution was mixed 200 μL DNS reagent, and the mixture was then incubated at 100 °C for 10 min. After that, 900 μL of DI water was added, and the absorbance at 540 nm (A540) was measured using a SpectraMax M3 microplate reader (Molecular Devices). The A540 values of different D-fructose concentrations (0-1.2 mg mL$^{-1}$) reacted with DNS reagent were measured to calculate the standard curve.

### Preparation of INU-NH$_2$

The approach employed in our research was modified from Ziegler-Borowska et al.[44]. In order to activate INU, weigh 3 g of inulinase dissolved in phosphate buffer solution (10 mM, pH 5.6), and 1 mg/mL INU solution (15 mL) was obtained by ultrafiltration based on the Bradford method. 20 mg BYTH was added to the INU solution, and 3 mL activator (50 mg EDC and 10 mg S-NHS) was also added to the INU solution to react at room temperature for 2 h. Subsequently, phosphate buffer (10 mM, pH 7.5) was added to the INU solution to adjust the pH of INU solution to pH 7.0 and incubated for 2 h. The INU-NH$_2$ product was obtained through the process of ultrafiltration followed by freeze-drying for future use.

### Synthesis of enzyme&cell@NKCOF-141

Typically, the cell (40 mg, 2.04 × 10$^5$ CFU/mg) was mixed and suspended in PBS (15 mL), which contains 3.3 mg TB and 17 mg INU-NH$_2$. Subsequently, the above solution was introduced into 5 mL PBS containing 10 mg of dissolved BYTH. The reaction mixture underwent a reaction for a duration of 2 h at room temperature after adding acetic acid (20 μL, 17.5 mM). The product was collected via centrifugation and subsequently subjected to PBS to produce enzyme&cell@NKCOF-141 after washing.

### Labeling on cells using propidium iodide (PI)

The PI storage solution (1 mg/mL) was prepared in sterile water and used at a working concentration of 50 μg mL$^{-1}$. Since PI could only label cells with impaired membrane integrity, a 20% isopropyl alcohol solution was integrated into the labeling process to increase cellular permeability, enabling comprehensive cell staining.

### Preparation of E-NH$_2$

After modifying the above method, the cells were activated to covalently immobilize NKCOF-141. Briefly, 2 mL PBS buffer (10 mM, pH 7.4) was used to dissolve 5 mg EDC and 1 mg S-NHS. Then, 2 mg BYTH and *E. coli* (8 mg, wet weight) were added to the solution and reacted at room temperature for 30 min. Subsequently, the NH$_2$-functionalized *E. coli* (E-NH$_2$) was collected by centrifugation and washed with PBS for two times.

### Typical protocol for the cascade reaction in enzyme&cell@NKCOF-141

The typical enzyme-cell biocatalytic cascade assay was conducted in 2 mL PB buffer (50 mM, pH 6.5) containing 10 g L$^{-1}$ inulin, 6 mg INU-NH$_2$&E-NH$_2$@NKCOF-141 at 50 °C for 10 min. Equal amounts of the free

enzyme&cell (INU-NH$_2$ and E-NH$_2$) were used instead of INU-NH$_2$&E-NH$_2$@NKCOF-141 as a control. The cascade reaction was started by the addition of inulin. The rate of the enzyme-cell cascade reaction was analyzed by HPLC for the production of D-allulose.

## Measurement of thermal inactivation

During incubation of INU-NH$_2$&E-NH$_2$ and INU-NH$_2$&E-NH$_2$@NKCOF-141 in pH 6.5 at 50 °C, samples were taken for cascade activity assay. The half-life ($t_{1/2}$) is defined as the time it takes for the cascade activity to decrease to half of its initial value. To determine the inactivation rate constant ($k_d$), the residual activity (%) was graphed against time in semi-logarithmic from Eq. (3).

$$\ln(A_t/A_0 \times 100) = -k_d t \tag{3}$$

where $A_0$ is the initial cascade activity and $A_t$ is the residual activity at a specific time. The examination of the half-lives for both free and immobilized forms was carried out using the first-order inactivation kinetic model. The formula is as follows:

$$t_{1/2} = \ln(2)/k_d \tag{4}$$

## Stability and recyclability tests

To evaluate stability, the free INU-NH$_2$&E-NH$_2$ and INU-NH$_2$&E-NH$_2$@NKCOF-141 were treated by incubation at heat (60°C) for 20 min, at methanol aqueous solution (50%) for 40 min, at ethanol aqueous solution (40%) for 30 min, and in Chymotrypsin (15 mg mL$^{-1}$) for 30 min, followed by cascade catalytic reactions to identify the remaining cascade activity. In the stability experiments, the cascade activity of untreated was recorded as 100% for each experimental set. The recyclability of INU-NH$_2$&E-NH$_2$@NKCOF-141 was evaluated in 50 mM PB buffer (pH 6.5) containing 10 mg mL$^{-1}$ inulin, and 6 mg mL$^{-1}$ co-immobilized INU-NH$_2$&E-NH$_2$@NKCOF-141, at 50 °C for 10 min. When the reaction was over, INU-NH$_2$&E-NH$_2$@NKCOF-141 was collected by centrifugation at 6000 × g. The content of D-allulose in the supernatant was detected by HPLC, and the content of D-allulose generated in the first reaction was recorded as 100%. The same substrate was added to the precipitate again to start the next round of reaction. After this round of reaction, the same centrifugation treatment was performed to measure the tagatose concentration in the supernatant and to compare the tagatose concentration with the first reaction further. The precipitate was used again for the next round of reactions.

## Enzyme-cell cascade catalysis in continuous-flow reaction system

The column reactor (with an inner diameter of 1 cm) was packed with INU-NH$_2$&E-NH$_2$@NKCOF-141, and the 300 mesh screen was installed at both ends of the column (4 mL in volume). 10 mg mL$^{-1}$ inulin solution was prepared through pH 6.5 PB buffer, which was pumped into the column reactor at a specific flow rate of 6.0 mL h$^{-1}$. Inulin solution passed through the bottom of the column reactor and out the top end, and the outflow was periodically collected for analysis using HPLC.

The space-time-yield (STY) was regarded as the mass of D-allulose produced per milliliter of INU-NH$_2$&E-NH$_2$@NKCOF-141 per day. The formula is as follows:

$$STY = \text{D-allulose (g)/Volume of INU-NH}_2\text{&E-NH}_2\text{@NKCOF-141(L)} \\ /\text{Reaction time (day)} \tag{5}$$

## Reporting summary

Further information on research design is available in the Nature Portfolio Reporting Summary linked to this article.

## Data availability

All data supporting the plots of this study are provided in this paper and Supplementary Information file, or available from the corresponding authors upon request. Source data are provided with this paper.

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

## Acknowledgements

The authors acknowledge the financial support from the National Key Research and Development Program of China (2021YFC2102100), the National Natural Science Foundation of China (22022808), Haihe Laboratory of Synthetic Biology (22HHSWSS00008) and CAS Youth Interdisciplinary Team program.

## Author contributions

Y.C. initiated and directed the project; D.Z., Y.Z. synthesized covalent organic frameworks (COFs) incorporating acylhydrazone linkages and collected, and analyzed the results; D.Z., J.T. synthesized the whole-cells@COFs, enzymes@COFs, and enzymes&whole-cells@COFs, and collected the data from powder X-ray diffraction, gas adsorption, CLSM, TEM, and the activity under the supervision of Y.C. and Z.Z., and D.Z. wrote the paper and H.H. give important comments on writing and all authors contributed to revising the paper.

## Competing interests

The authors declare no competing interests.
