## [Peer Review File · Nature Communications]

Co-immobilization of whole cells and enzymes by covalent organic framework for biocatalysis process intensificationReviewers' Comments:

Reviewer #1:

Remarks to the Author:

The authors present an interesting study in which they co-immobilize enzymes (unlinase) and whole cells (*E. coli*) for the conversion of inulin to D-allulose. The co-immobilization of enzymes and whole-cell by porous Covalent-Organic Frameworks is particularly interesting. This aspect as well as the novelty of this work has the potential to be of interest for the readership of Nature Communications. However, prior to recommendation for publication, a major clarification is needed.

I am very confused by the results obtained using propidium iodide (PI) to stain *E. coli*. The authors use PI to stain the bacteria to assess their localization within the material by CLSM (eg, Fig 4). PI is typically and widely used in live/dead assay as an indicator of dead bacteria. PI is a fluorescent intercalating agent that is not membrane-permeable. Maybe I'm missing an information but to me the authors are imaging damaged cells.

The fact that the encapsulation could damage some bacteria membrane is not surprising and has been previously demonstrated with many materials. The reported material is exhibiting some catalytic activity, suggesting that either there are some undamaged cells or that the catalytic activity results from released extracellular enzymes. This should be clarified prior to publication.

To partly answer the question, I believe strongly improving the characterization of the material is thus required:

- Is there some bacteria non-stained by PI visible by optical microscopy?
- If so what is their ratio compared to damaged cells? (ie performing live/dead assay)
- Are they coated by the COF? Are all cells coated by the COF?
- Is the COF only formed around the cells/enzyme or also in the bulk? The authors provided TEM images of single cell at high magnification (note that the same picture is given in Figure 2 and S5, I believe the authors did not image just on bacteria and could provide different images), lower magnification should be provided to assess the homogeneity of the sample.
- Is the coating uniform and what is the coating thickness (TEM cross section)?

Also, concerning the *E. coli*:

- Can the *E. coli* form colony after immobilization?
- The amount of bacteria is given in dry weight, typically cfu are used to describe their amount. What are the cfu introduced in the synthesis?
- For catalytic activity the authors state that the same weight of cells is introduced between free and immobilized bacteria. Similarly, I believe comparing identical cfu amount would be more accurate.

Also, I would suggest addressing several comments:

- In FT-IR spectra of the E@COF (fig 2d and S12), the authors attribute the band at 1660 cm⁻¹ to a shift in the vibration band of the amide I of *E. coli*, but this vibration band could also result from the vibration band of the COF, no? This observation leads the authors to conclude that a strong interaction is occurring between the COF and the cell surface. TEM cross-section to highlight the interface between the COF and the bacteria should be performed to validate this assumption. (in line with my previous comment)
- The authors use the Bradford assay to quantify the non-immobilized INU and deduce the loading efficiency and capacity. However, as stated in the Methods and displayed in figure S18, the protein BSA is used for the calibration curve. The Bradford assay relies mainly on the binding of basic and aromatic amino acid and thus depends on the amino acid sequence of the protein. The calibration curve should thus be created with INU, and values adjusted accordingly.
- INU and *E. coli* amino functionalisation. FT-IR spectroscopy is used to assess the functionalization of INU and *E. coli* with BYTH. I am not convinced that the very slight increase in intensity of the vibration band around 3400 cm⁻¹, especially in the case of *E. coli*, is sufficient to indicate that there is some grafting. It may be hard to undoubtedly evidence and the results clearly prove the protocol favors the encapsulation, the authors should however either provide further evidence for the formation of BYTH-modified *E. coli* or adjust their statement.
- Co-immobilization of the enzyme and bacteria. A control experiment of CLSM without COF, to check

if without COF the same enzyme-bacteria assemblies are observed would be good.

- The catalytic production is preserved even after 14 days. I am wondering what is a free E. coli cell life span? Does that mean the cells can divide and reproduce within the column?

Some minor comments:

- There is a numbering mismatch with Fig S6 (PXRD) that is cited in the main text as cell activity.
- Fig 3a is, in my opinion, misleading. The size of the INU enzyme compared to the size of the NKCOF-141 pores doesn't seem in agreement with the stated values.
- Fig S23a, has a labelling typo, enzyme should be cell.
- Fig 4e, for comparison sake, it would be good to add the INU-NH₂ + E-NH₂ system.

Reviewer #2:

Remarks to the Author:

This is a very interesting article that deals with the design of covalent organic frameworks (COFs) to co-immobilize whole cells and enzymes. The article is generally well written. The co-immobilization enzyme@COFs are well characterized by combining multiple techniques, show relatively high cascade activity, and demonstrate excellent universality. This synthesis strategy formed enzyme@COFs can be produced in gram scale and conducts continuous-flow reaction. This article provides novel and interesting results for co-immobilization of whole cells and enzymes. This article should be accepted after considering these considerations:

- 1) Some figures are too small and it makes the reading of the paper difficult. As an example, Figure 2c is too small and the authors should make some adjustments.
- 2) For enzyme pre-incubation with monomer BYTH to increase loading (Fig. 3c), is this strategy universal, e.g. other enzymes.
- 3) A higher cascade activity was observed for INU-NH₂@NKCOF-141 in comparison to other matrices (calcium alginate, polyvinyl alcohol, and ZIF-8) but the loading of those matrices on the enzyme and the cell is different. The authors should specify it in the article. Normally, the cascade activity of biohybrid materials should be compared with a similar catalyst content. When making comparisons are the enzyme and cell loading similar in those materials?
- 4) In addition to CLSM, the TEM images of INU-NH₂@NKCOF-141 with high magnification should be given to compare with INU-NH₂@NKCOF-141.
- 5) The graphic and notes of Supplementary Fig. 23 are mismatched?

Reviewer #3:

Remarks to the Author:

The manuscript of Zheng et al. develops a new co-immobilization platform for integrating enzymes and cells in a simplified and efficient way. The platform utilizes covalent organic frameworks (COFs) to immobilize enzymes uniformly, providing high efficiency, stability, and recyclability. The process allows for the fabrication of enzyme-cell biocatalysts on a gram-scale, enabling continuous-flow devices for bioconversion of natural products. I believe that the experimental design is sound and only have a few comments for clarification that need to be addressed in the manuscript.

1. How does the homogeneity of the COF coating on cells and enzymes? It would be beneficial to provide a larger-scale TEM image of the E-NH₂@INU-NH₂@NKCOF-141 co-immobilization structure to gain a more comprehensive understanding.
2. In this study, COF serves as a carrier for immobilizing cells and enzymes. It is crucial to investigate whether the thickness of the COF layer impacts the reactivity of the enzyme@NKCOF-141 complex. Therefore, the author should provide data on various thicknesses of the COF carrier and conduct a comparative analysis of their protective efficiencies.
3. How does the author regulate the input of COF monomers to ensure that the COF can be coated around the cell surface instead of forming COF composite individually?

4. As the author mentioned in the manuscript, both NKCOF-98 and COF-421-B can be coated on the surface of cells. Is there a difference in their ability to load enzymes?
5. How does the author regulate the loading amount of INU?
6. The significant difference in catalytic efficiency when using different materials to coat cells in Figure 4e suggests that the authors should provide the amount of enzyme loaded for each material.
7. In figure 6c, how is the activity of *E. coli* ensured during such a long catalytic process?

Reviewer #1:

The authors present an interesting study in which they co-immobilize enzymes (unlinase) and whole cells (*E. coli*) for the conversion of inulin to D-allulose. The co-immobilization of enzymes and whole-cell by porous Covalent-Organic Frameworks is particularly interesting. This aspect as well as the novelty of this work has the potential to be of interest for the readership of Nature Communications. However, prior to recommendation for publication, a major clarification is needed.

Response: We thank the reviewer's high comments.

Comments 1: I am very confused by the results obtained using propidium iodide (PI) to stain *E. coli*. The authors use PI to stain the bacteria to assess their localization within the material by CLSM (eg, Fig 4). PI is typically and widely used in live/dead assay as an indicator of dead bacteria. PI is a fluorescent intercalating agent that is not membrane-permeable. Maybe I'm missing an information but to me the authors are imaging damaged cells.

The fact that the encapsulation could damage some bacteria membrane is not surprising and has been previously demonstrated with many materials. The reported material is exhibiting some catalytic activity, suggesting that either there are some undamaged cells or that the catalytic activity results from released extracellular enzymes. This should be clarified prior to publication.

To partly answer the question, I believe strongly improving the characterization of the material is thus required:

- Is there some bacteria non-stained by PI visible by optical microscopy?
- If so what is their ratio compared to damaged cells? (ie performing live/dead assay)

Response: We thank the reviewer for the comment and suggestion. Originally, in order to visualize the localization of the enzyme and the cells after co-immobilization by NKCOF-141, we used PI to label the cells, and utilized a staining solution containing 20% isopropanol to facilitate the alteration of cell permeability, enabling comprehensive labeling of the cells. Using this method, no bacteria non-stained by PI were visible by optical microscopy, and all cells were labeled in red (Supplementary Fig. 26).

Per the reviewer' suggestion, we utilized the nucleic acid stain SYTO 9 to label all cells (live /dead cells can be labeled) and co-immobilized the labeled cells and Rhodamine B-labeled INU-NH₂ (RhB-INU-NH₂) by NKCOF-141 to further improve the characterization. The result demonstrated that enzyme (RhB-INU-NH₂) was uniformly immobilized on the cell surface by NKCOF-141 (Figure R1), showing the same results as FITC-INU-NH₂&E-NH₂@NKCOF-141 (label the cells with PI).

Figure R1. CLSM images of RhB-INU-NH₂&E-NH₂@NKCOF-141: (i) optical image; (ii) excitation at 540 nm and tracking the fluorescence within a spectral range of 560 to 600 nm; (iii) excitation at 488 nm and tracking the fluorescence within a spectral range of 480 to 540 nm; (iv) overlap image of three images. The cell was stained with SYTO 9 (green). Scale bar: 2 μm.

Per the reviewer' suggestion, we also explored the biocompatibility of the immobilization process by live/dead assay to further improve the characterization (Supplementary Fig. 9a, b). The two figures show that the ratio of living cells (green) and dead cells (red) among E@NKCOF-141 was similar to the free E, with only a few dead cells in red. This indicated that NKCOF-141 was biocompatible with minimized cell destruction. In addition, although E@NKCOF-141 showed a delayed growth curve compared to free *E. coli* cells, E@NKCOF-141 exhibited consistent growth viability with free E (Supplementary Fig. 9c). These findings highlighted that the immobilization of E by NKCOF-141 was mild enough to allow cell division even after the coating procedure.

Comments 2:

- Are they coated by the COF? Are all cells coated by the COF ?
- Is the COF only formed around the cells/enzyme or also in the bulk? The authors provided TEM images of single cell at high magnification (note that the same picture is given in Figure 2 and S5, I believe the authors did not

image just on bacteria and could provide different images), lower magnification should be provided to assess the homogeneity of the sample.

- Is the coating uniform and what is the coating thickness (TEM cross section) ?

Response: We thank the reviewer for the comments. Immobilization of cells by COF was related to monomer concentrations (TB and BYTH). Along with the increase in concentration of TB and BYTH, the encapsulation efficiency of the cells will be improved. COF was formed in the bulk when too much of the monomer was added to the solution. Therefore, we chose a slight excess of monomer content to achieve complete immobilization of cells, while only a small amount of COFs without coating cells exist.

Per the reviewer's suggestion, lower magnification has been added to Supplementary Fig. 31b. CLSM (Supplementary Fig. 26 and Figure R1) and TEM (Supplementary Fig. 31) results demonstrated that INU-NH₂@NKCOF-141 were uniformly wrapped around all cells, and there was only a little excess of COFs without coating the cells. In addition, the result of TEM indicated that the average thickness was around 200 nm (Figure R2).

Figure R2. The thickness measurement of INU-NH₂@E-NH₂@NKCOF-141.

According to the suggestion, the TEM images of E@NKCOF-141 under lower magnification have been added to Supplementary Fig. 5c, and it shows the homogeneous coating of E by NKCOF-141. The coating of NKCOF-141 was also clearly observed in the TEM

micrograph of the microtomed E@NKCOF-141, and the average thickness was around 20 nm (Supplementary Fig. 6).

Comments 3: Also, concerning the *E. coli*: 15.30

- Can the *E. coli* form colony after immobilization?
- The amount of bacteria is given in dry weight, typically cfu are used to describe their amount. What are the cfu introduced in the synthesis ?
- For catalytic activity the authors state that the same weight of cells is introduced between free and immobilized bacteria. Similarly, I believe comparing identical cfu amount would be more accurate.

Response: We thank the reviewer for the comment and suggestion. Yes, the *E. coli* could form colony after immobilization. As shown in Supplementary Fig. 9c, we demonstrated that E@NKCOF-141 was able to divide and reproduce in a liquid medium. Per the reviewer's suggestion, we also explored the growth characteristics of E@NKCOF-141 when cultured on solid agar plates. The colonies of E@NKCOF-141 on plates revealed that *E. coli* could form colony after immobilization (Figure R3).

Figure R3. The plate colonies of E@NKCOF-141 under the dilution multiples.

In order to calculate colony-forming units (CFU) of bacteria during immobilization (2 mg/mL), The bacteria were quantified using the ten-fold serial dilutions method and the plates were incubated overnight. CFU was determined by counting the number of colonies on each plate and using dilution calculations to calculate the CFU of the original bacterial concentrations (Figure R4). The mean bacterial count in the

immobilized system was 2.04×10^5 CFU/mg (Table 1). The corresponding changes have been highlighted in yellow in the main text and supporting Information.

Per the suggestion, we compared the activity at the same number of free and immobilized cells (Supplementary Fig. 7b). The corresponding changes have been highlighted in yellow in the manuscript and supporting Information.

Figure R4. The plate colonies of free E under the dilution multiples. The blank represents the used culture medium without bacterial inoculation.

Table 1. The number of free E at various dilution ratios.

Dilution factor	10^1	10^2	10^3
Colonies counted	Too much to count	214, 194	29, 24

Comments 4: - In FT-IR spectra of the E@COF (fig 2d and S12), the authors attribute the band at 1660 cm^{-1} to a shift in the vibration band of the amide 1 of E. coli, but this vibration band could also result from the vibration band of the COF, no? This observation lead the authors to conclude that a strong interaction is occurring between the COF and the cell surface. TEM cross-

section to highlight the interface between the COF and the bacteria should be performed to validate this assumption. (in line with my previous comment)
Response: We thank the reviewer for the comment. No, the band at 1660 cm⁻¹ resulted from the interaction of COF with the cell surface. We tested the FT-IR by simply mixing cells and NKCOF-141. The result (Figure R5) demonstrated that simple mixing of COF (1670 cm⁻¹) with the cells (COF not interacting with the cell) did not produce a shift in the 1633 cm⁻¹ characteristic peak, whereas the immobilization of cells by COF would have a shift from 1633 cm⁻¹ to 1660 cm⁻¹ (Fig. 2d).

Per the suggestion, we added the data of TEM images of microtomed E@NKCOF-141, which further illustrated the immobilization of NKCOF-141 on the cell surface (Supplementary Fig. 6). The corresponding changes have been highlighted in yellow in supporting Information.

Figure R5. FT-IR spectra of the E, NKCOF-141 and mixing of E and NKCOF-141.

Comments 5: The authors use the Bradford assay to quantify the non-immobilized INU and deduce the loading efficiency and capacity. However, as stated in the Methods and displayed in figure S18, the protein BSA is used for the calibration curve. The Bradford assay relies mainly on the binding of basic and aromatic amino acid and thus depends on the amino acid sequence of the protein. The calibration curve should thus be created with INU, and values adjusted accordingly.

Response: We thank the reviewer for the suggestion. Per your suggestion, we have constructed the standard curve for INU and employed this updated curve to calculate the loading capacity of INU (Supplementary Fig. 19a). The corresponding changes have been highlighted in yellow in the main text and supporting Information.

Comments 6: - INU and *E. coli* amino functionalization. FT-IR spectroscopy is used to assess the functionalization of INU and *E. coli* with BYTH. I am not convinced that the very slight increase in intensity of the vibration band around 3400 cm⁻¹, especially in the case of *E. coli*, is sufficient to indicate that there is some grafting. It may be hard to undoubtedly evidence and the results clearly prove the protocol favors the encapsulation, the authors should however either provide further evidence for the formation of BYTH-modified *E. coli* or adjust their statement.

Response: We thank the reviewer for the suggestion. In the field of enzyme immobilization, the activation of carboxyl groups (-COOH) and their coupling with amino groups (-NH₂) using EDC and NHS was a commonly used and effective strategy (*Angew. Chem. Int. Ed.* 2018, 57, 16754; *Mater. Chem. Front.* 2021, 5, 3859; *ACS Appl. Mater. Interfaces* 2022, 14, 2881; *Anal. Chim. Acta.* 2020, 1140, 228). We have adjusted our statement about BYTH-modified *E. coli* as suggested. The corresponding changes have been highlighted in yellow in the manuscript.

Comments 7: -Co-immobilization of the enzyme and bacteria. A control experiment of CLSM without COF, to check if without COF the same enzyme-bacteria assemblies are observed would be good.

Response: We thank the reviewer for the suggestion. Per the suggestion, we have tried to incubate the enzyme with the cells. The treatment of the enzyme and the cells was consistent with the operation of the co-immobilization system, but the solution did not contain TB, and thus, no COFs were formed. As shown in Supplementary Fig. 27, we found that enzymes did not bind on the cell surface to form enzyme-bacteria assemblies in the absence of COF. The corresponding

changes have been highlighted in yellow in the main text and supporting Information.

Comments 8:-The catalytic production is preserved even after 14 days. I am wondering what is a free *E. coli* cell life span? Does that mean the cells can divide and reproduce within the column?

Response: We thank the reviewer for the comment. *E. coli* cells are capable of maintaining their lifespan for up to a month or even several months. In the absence of specific essential nutrients, microorganisms entered a state of starvation or dormancy, ceasing their growth and reproduction. These dormant cells, commonly known as resting cells, possessed the capability to endure for extended periods, such as surviving in sterile stream water for up to 234 days and in sterile soil for as long as 179 days (*Int. J. Microbiol.* 2011, 340506). Furthermore, in some natural environments, the survival times of *E. coli* in bovine manure extended up to 21 months (*Crit. Rev. Microbiol.* 2015, 41, 273), and *E. coli* survived for more than a month in soils from Salinas Valley, California (*Environ. Sci. Pollut. Res.* 2021, 28, 5575). And under aqueous conditions, *E. coli* cells could survive for over a month in lake (*J. Water Health* 2006, 4, 389) and river environments (*J. Appl. Microbiol.* 2004, 5, 922).

Cells could not divide and reproduce within the column. During the continuous-flow reaction, the solution only contained 10 mg mL⁻¹ of inulin (sugar, C sources) and PB buffer (containing Na⁺ and PO₄⁻), lacking essential nitrogen sources, inorganic salts (K⁺, Mg²⁺), and vitamins required for cell growth and reproduction. When cells lack certain essential nutrients, they enter a state of starvation or dormancy, ceasing growth and reproduction. As a result, the cells did not divide and reproduce within the column. However, as these cells (resting cells) contain various enzyme systems, they retain catalytic activity. Additionally, we inoculated cells into various sterilized solutions and incubated them at 30 °C for 12 hours (Figure R6). We only observed growth of cells in LB medium; however, no growth or reproduction of cells was observed in solutions lacking essential nutrients (50 mM PB,

50 mM PB containing inulin, and 50 mM PB containing inulin and INU-NH₂).

Figure R6. The growth of cells in different solutions.

Comments 9:-Some minor comments:

- There is a numbering mismatch with Fig S6 (PXRD) that is cited in the main text as cell activity.
- Fig 3a is, in my opinion, misleading. The size of the INU enzyme compared to the size of the NKCOF-141 pores doesn't seem in agreement with the stated values.
- Fig S23a, has a labelling typo, enzyme should be cell.
- Fig 4e, for comparison sake, it would be good to add the INU-NH₂ + E-NH₂ system.

Response: We thank the reviewer for the comment and suggestion. Per the suggestion, the mismatched images have been corrected (Supplementary Fig. 7). According to the suggestion, we have made adjustments to the size of the INU enzyme and NKCOF-141 pores (Fig. 3a).

In addition, we have proofread the manuscript and corrected the labelling typo (Supplementary Fig. 23). The suggested modification (The INU-NH₂ + E-NH₂ system) has been added to Figure 4e. The corresponding changes have been highlighted in yellow in the main text.

Reviewer #2:

This is a very interesting article that deals with the design of covalent organic frameworks (COFs) to co-immobilized whole cells and enzymes. The article is generally well written. The co-immobilization enzyme&cell@COFs are well characterized by combining multiple techniques, show relatively high cascade activity, and demonstrate excellent universality. This synthesis strategy formed enzyme&cell@COFs can be produced in gram scale and conducts continuous-flow reaction. This article provides novel and interesting results for co-immobilization of whole cells and enzymes.

Response: We appreciate the reviewer's high comments and support of our work.

This article should be accepted after considering these considerations:

Response: We thank the reviewer for the suggestion, and we have revised the manuscript point by point as follows.

Comments 1: Some figures are too small and it makes the reading of the paper difficult. As an example, Figure 2c is too small and the authors should make some adjustments.

Response: We thank the reviewer for the suggestion. Per the suggestions, we have modified the size of the images and notes in the main manuscript (Fig. 2c). And the corresponding changes have been highlighted in yellow in the main text.

Comments 2: For enzyme pre-incubation with monomer BYTH to increase loading (Fig. 3c), is this strategy universal, e.g. other enzymes.

Response: We thank the reviewer for the comment. Per the suggestions, we chose Glucose oxidase (GOx) as the other example to study BYTH with modification to increase loading. The encapsulation loading was determined via a standard Bradford assay method (the calibration curves were created using BSA, Figure R7). By immobilizing GOx through NKCOF-141, the loading capacity of GOx@NKCOF-141 was only 0.036 g g⁻¹. The low loading of GOx was consistent with the results of low terminal amino groups (-NH₂) of GOx (Table 2).

Figure R7. Corresponding standard calibration curve of BSA based on Bradford assay.

Table 2. Amino acid composition analysis of GOx from *Aspergillus niger* (PDB: 1GAL).

Amino acid	Number	% mol/mol
Arginase	22	3.77
Lysine	15	2.57
Aspartic acid	36	6.17
Glutamic acid	30	5.15

Thereby, we pre-incubated GOx with BYTH under EDC/s-NHS conditions to obtain GOx-NH₂. The loadings of GOx-NH₂@NKCOF-141 to GOx were significantly increased to 0.294 g g⁻¹ (Figure R8). This result demonstrated that the pre-incubation strategy was effective in increasing the loading of COFs for enzyme immobilization and that the strategy was universal.

Figure R8. The loading amounts of Gox@NKCOF-141 and GOx-NH₂@NKCOF-141.

Comments 3: A higher cascade activity was observed for INU-NH₂&E-NH₂@NKCOF-141 in comparison to other matrices (calcium alginate, polyvinyl alcohol, and ZIF-8) but the loading of those matrices on the enzyme and the cell is different. The authors should specify it in the article. Normally, the cascade activity of biohybrid materials should be compared with a similar catalyst content. When making comparisons are the enzyme and cell loading similar in those materials?

Response: We thank the reviewer for the comment and suggestion. When comparing the catalytic activities of different enzyme-cell co-catalyst complexes, the enzyme and cell loadings in these materials were consistent, with a loaded enzyme quantity of 1 mg and a cell quantity of 4 mg. The loading of the four materials on cells was nearly 100%, and the loading of the enzyme varied (Table 3). Therefore, the immobilized system was adjusted to achieve an enzyme loading of 1 mg under the condition of 4 mg cell addition to ensure the same amount of enzyme and cells in each material. The corresponding changes have been highlighted in yellow in the main text.

Table 3. The loading of INU-NH₂ on different co-immobilized materials

Materials	LE (%)	LC (g g ⁻¹)
Calcium alginate	39%±4%	0.21
Polyvinyl alcohol	42%±3%	0.09
ZIF-8	86%±3%	0.24
NKCOF-141	95%±2%	0.65

Comments 4: In addition to CLSM, the TEM images of INU-NH₂&E@NKCOF-141 with high magnification should be given to compare with INU-NH₂&E-NH₂@NKCOF-141.

Response: We thank the reviewer for the comment. Per the suggestion, we have added the TEM image of INU-NH₂&E@NKCOF-141 (Supplementary Fig. 24). The corresponding changes have been highlighted in yellow in the manuscript and supporting Information.

Comments 5: The graphic and notes of Supplementary Fig. 23 are mismatched?

Response: We thank the reviewer for the suggestion, and we have proofread the manuscript and corrected the labelling typo (Supplementary Fig. 23). The corresponding changes have been highlighted in yellow in Supporting Information.

Reviewer #3:

The manuscript of Zheng et al. develops a new co-immobilization platform for integrating enzymes and cells in a simplified and efficient way. The platform utilizes covalent organic frameworks (COFs) to immobilize enzymes uniformly, providing high efficiency, stability, and recyclability. The process allows for the fabrication of enzyme-cell biocatalysts on a gram-scale, enabling continuous-flow devices for bioconversion of natural products. I believe that the experimental design is sound and only have a few comments for clarification that need to be addressed in the manuscript.

Response: We appreciate the reviewer's high comments and support of our work.

Comments 1: How does the homogeneity of the COF coating on cells and enzymes? It would be beneficial to provide a larger-scale TEM image of the E-NH₂&INU-NH₂@NKCOF-141 co-immobilization structure to gain a more comprehensive understanding.

Response: We thank the reviewer for the comment and suggestion. Per the suggestion, we have added TEM image of the E-NH₂&INU-NH₂@NKCOF-141 with a larger scale of view. The results demonstrated a uniform encapsulation of cells by COF particles (Supplementary Fig. 31), while enzymes also exhibit homogeneous encapsulation within COFs, as observed by confocal laser scanning microscopy (Supplementary Fig. 26). Corresponding changes have been added and highlighted in yellow in the Supporting Information.

Comments 2: In this study, COF serves as a carrier for immobilizing cells and enzymes. It is crucial to investigate whether the thickness of the COF layer impacts the reactivity of the enzyme&cell@NKCOF-141 complex. Therefore, the author should provide data on various thicknesses of the COF carrier and conduct a comparative analysis of their protective efficiencies.

Response: We thank the reviewer for the suggestion. Per the suggestion, we have tested the thickness of the COF coatings of enzyme&cell@NKCOF-141 at different synthesis times (Supplementary Fig. 36a, b). The average thicknesses of the COF coatings were 123 nm, 157 nm, and 197 nm under 30, 60, and 120 min synthesis conditions, respectively. In addition, the protective efficiencies were enhanced as the integrality and thickness of the shell increased, which was consistent with previous reports (Supplementary Fig. 36c) (*ACS Nano* 2021, 15, 15920). The corresponding changes have been highlighted in yellow in the manuscript and supporting Information.

Comments 3: How does the author regulate the input of COF monomers to ensure that the COF can be coated around the cell surface instead of forming COF composite individually?

Response: We thank the reviewer for the comment. During the self-assembly process of COF, it was inevitable that individual COF particles would form independently. However, we could address this issue by pre-incubating bacteria in an acidic TB solution for 20-30 minutes to allow TB to accumulate on the cell surface. Subsequently, BYTH solution was added. At this point, BYTH is easily assembled with the highly enriched TB on the cell surface, effectively forming a COF coating on the cell surface. Detailed operating instructions have been added in the manuscript (highlighted in yellow).

Comments 4: As the author mentioned in the manuscript, both NKCOF-98 and COF-421-B can be coated on the surface of cells. Is there a difference in their ability to load enzymes?

Response: We thank the reviewer for the suggestion. Per the suggestion, we have tested the loading amount and efficiency of NKCOF-98 and COF-42-B for encapsulation of various enzyme (INU-

NH₂) concentrations (0.5-1.5 mg/mL). The encapsulation efficiency and loading capacities were determined via a standard Bradford assay method (the calibration curves were created using BSA). In the system of INU-NH₂@NKCOF-98 (Figure R9a), when the enzyme addition amount increased from 0.5 to 1.0 mg/mL, the loading capacity showed an increasing trend, reaching a maximum loading of 0.154 g/g. Continuing to increase the enzyme amount did not lead to a significant change in the loading capacity, indicating that the loading of INU-NH₂ by NKCOF-98 had reached saturation at 1.0 mg/mL. In the system of INU-NH₂@COF-42-B, the loading capacity consistently increased as the enzyme concentration increased from 0.5 to 1.5 mg/mL, reaching the maximum loading at 1.5 mg/mL, which was 0.094 g/g (Figure R9b).

Figure R9. (a) Loading amount for INU-NH₂@NKCOF-98 and (b) INU-NH₂@COF-42-B.

Comments 5: How does the author regulate the loading amount of INU?

Response: We thank the reviewer for the comment. To control the amount of loaded enzyme, the encapsulation of INU-NH₂ was carried out with varying enzyme concentrations ranging from 0.5 to 1.5 mg mL⁻¹. Per the suggestion, we have tested the loading amount at different enzyme concentrations. As shown in Figure R10, the rise in enzyme addition from 0.5 to 1.25 mg mL⁻¹ led to an increase in loading capacity, peaking at 0.257 g/g. However, a further increment in enzyme concentration resulted in a decrease in loading amount. This phenomenon may be attributed to the excessive enzyme hindering the

synthesis process of NKCOF-141, thus reducing material synthesis and leading to a significant drop in loading capacity.

Figure R10. Loading amount for INU-NH₂@NKCOF-141.

Comments 6: The significant difference in catalytic efficiency when using different materials to coat cells in Figure 4e suggests that the authors should provide the amount of enzyme loaded for each material.

Response: We thank the reviewer for the comment and suggestion. Per the suggestions, we have tested the loading of INU-NH₂ on different co-immobilized materials using the Bradford assay (Table 3). When comparing the catalytic activities of different enzyme-cell co-catalyst complexes, the enzyme and cell loads in these materials were consistent, with a loaded enzyme quantity of 1 mg and a cell quantity of 4 mg. And the corresponding changes have been highlighted in yellow in the main text.

Table 3. The loading of INU-NH₂ on different co-immobilized materials

Materials	LE (%)	LC (g g ⁻¹)
Calcium alginate	39%±4%	0.21
Polyvinyl alcohol	42%±3%	0.09
ZIF-8	86%±3%	0.24
NKCOF-141	95%±2%	0.65

Comments 7: In figure 6c, how is the activity of E. coli ensured during such a long catalytic process?

Response: We thank the reviewer for the comment. Resting cells of *E. coli* have been reported in the literature to maintain cell viability for up to several months (Int. J. Microbiol. 2011, 340506). And, the immobilization of cells could further extend their lifetime to enhance their performance (Org. Biomol. Chem. 2015, 13, 10086). Yun et al. immobilized *E. coli* expressing endoinulinase of *Pseudomonas sp.* in alginate beads, and continuous production of inulo-oligosaccharide with immobilized cell for 15 days at 50°C without significant loss of activity (Bioproc. Biosyst. Eng. 1999, 21, 101). Gibello et al. immobilized *E. coli* expressing 3,4-dihydroxyphenylacetate 2,3-dioxygenase from *Klebsiella pneumoniae* in alginate beads. The immobilized recombinant cells were well maintained for at least 15 days in a glass column reactor (Curr. Microbiol. 2004, 49, 390). Jung et al. constructed recombinant *Escherichia coli* cells containing L-arabinose isomerase from *Geobacillus stearothermophilus* and captured the *E. coli* cells in alginate beads with a half-life of 34 days for the production of tagatose in a packed bed bioreactor (Biotechnol. Prog. 2015, 21, 1335). Kovalenko et al. immobilized *E. coli* BL21(DE3) cells expressing glucose isomerase from *Arthrobacter nicotianae* in SiO₂ xerogel. The immobilized biocatalyst had a half-life of inactivation of ~60 days under continuous hydrolysis conditions in a fixed-bed reactor (Appl. Biochem. Microbiol. 2011, 47, 151). All of these examples illustrated that immobilized *E. coli* resting cells maintained their catalytic activity over a long period of time.

Reviewers' Comments:

Reviewer #1:

Remarks to the Author:

The authors have clarified my initial confusion, and have thoroughly addressed the reviewers' concerns. I am glad to support the publication of this study, provided a few clarifications, mainly related to the enzyme content of the material are addressed.

The authors addressed my previous comment on the calibration curve for the Bradford assay, and performed a calibration curve using INU which appeared to be significantly different from the BSA calibration curve, suggesting the previous loading capacity in INU were underestimated. I have a few questions concerning this adjustment

1) The corrected loading capacity for INU-NH2@NKCOF-141 was $> 2\text{g.g}^{-1}$; this value sounds very high, the authors may want to double check the value.

2) The authors may also want to check the value for the loading efficiency of INU-NH2@NKCOF-141 in the manuscript, as the value remained the same than in the previous version.

3) The enzymatic activity % of immobilized INU-NH2 remained identical than in the previous version (fig. S22) where it was measured with a lower amount of enzyme within the material. I am wondering, if the authors did consider, in the revised version, the corrected amount of INU-NH2 within the material and performed the experiment with a different mass of material?

4) Similarly, for INU-NH2&E-NH2@NKCOF-141 activity in comparison to INU-NH2&E-NH2 (fig 4e), did the author considered the difference in the INU loading compared to what they had previously estimated?

5) Finally, from the answer to reviewer#2 (comments 3) and reviewer#3 (comments 6), it seems the loading of INU-NH2 within INU-NH2&E-NH2@NKCOF-141, is different than for INU-NH2@NKCOF-141, it may be relevant to include the loading of INU in INU-NH2&E-NH2@NKCOF-141 in the manuscript.

Reviewer #2:

Remarks to the Author:

This manuscript was improved well according to reviewers' comments and I agree with the acceptance.

Reviewer #3:

Remarks to the Author:

All my questions have been addressed.

Reviewer #1:

The authors have clarified my initial confusion, and have thoroughly addressed the reviewers' concerns. I am glad to support the publication of this study, provided a few clarifications, mainly related to the enzyme content of the material are addressed.

Response: We appreciate the reviewer's high comments and support of our work.

The authors addressed my previous comment on the calibration curve for the Bradford assay, and performed a calibration curve using INU which appeared to be significantly different from the BSA calibration curve, suggesting the previous loading capacity in INU were underestimated. I have a few questions concerning this adjustment

1) The corrected loading capacity for INU-NH₂@NKCOF-141 was > 2g.g⁻¹; this value sounds very high, the authors may want to double check the value.

Response: We thank the reviewer for the comment. Per the suggestions, we again checked the loading of INU-NH₂ in 20 mL immobilization system weighing 3.3 mg TB, 10 mg BYTH, and 30 mg INU-NH₂ (1.14 mg/mL, using the INU calibration curve), and the obtained INU-NH₂@NKCOF-141 possessed the same high enzyme loading (> 2g.g⁻¹, 2.278 g.g⁻¹).

2) The authors may also want to check the value for the loading efficiency of INU-NH₂@NKCOF-141 in the manuscript, as the value remained the same than in the previous version.

Response: We thank the reviewer for the comment. INU-NH₂ added to the immobilization system can be efficiently loaded by NKCOF-141, with nearly 100% loading efficiency for enzymes. Based on this, the value of the loading efficiency of INU-NH₂ was not affected by changes in the standard curve.

3) The enzymatic activity % of immobilized INU-NH₂ remained identical than in the previous version (fig. S22) where it was measured with a lower amount of enzyme within the material. I am wondering, if the authors did consider, in the revised version, the corrected amount of INU-NH₂ within the material and performed the experiment with a different mass of material?

Response: We thank the reviewer for the comment. We performed the experiment with the corrected amount of INU-NH₂ in the revised version. In the previous version, we weighed 30 mg of INU-NH₂ and dissolved it in 20 mL immobilization system to obtain an enzyme concentration of 0.1 mg/mL calculated from a BSA standard curve. And in the new version, the addition of 30 mg INU-NH₂ in 20 mL system was calculated to give an enzyme concentration of 1.14 mg/mL using the INU calibration curve. 30 mg INU-NH₂ achieved near 100% encapsulation in the immobilized system. Although the enzyme concentration and loading capacity calculated from the different standard curves were different, the amount of immobilized INU-NH₂ was actually the same (30 mg). Hence, the relative activity of immobilized INU-NH₂ was not affected by changes in the standard curve. The corresponding changes have been highlighted in yellow in the main text.

4) Similarly, for INU-NH₂&E-NH₂@NKCOF-141 activity in comparison to INU-NH₂&E-NH₂ (fig 4e), did the author considered the difference in the INU loading compared to what they had previously estimated?

Response: We thank the reviewer for the comment. We considered the difference in the INU loading by changes in the enzyme calibration curves, but the activity ratio was not affected by changes in the standard curve. In the previous version, we weighed 17 mg of INU-NH₂ and dissolved it in 20 mL co-immobilization system (0.05 mg mL⁻¹, calculated from the BSA calibration curve). In the new version, the description in the synthesis method was changed to directly weighing 17 mg of INU-NH₂, and the concentration of enzyme in the co-immobilized system was 0.62 mg mL⁻¹ calculated from the INU calibration curve. The added 17 mg INU-NH₂ achieved almost complete loading in the 20 mL immobilization system, and the immobilized 17 mg INU-NH₂ by NKCOF-141 had different loading capacities depending on the calibration curve. Whereas, under the condition that the amount of added enzyme was deterministic, the activity ratio of INU-NH₂&E-NH₂ to INU-NH₂&E-NH₂@NKCOF-141 containing the same amount of enzyme (17 mg) was not affected by changes in the enzyme calibration

curves. The corresponding changes have been highlighted in yellow in the main text.

5) Finally, from the answer to reviewer#2 (comments 3) and reviewer#3 (comments 6), it seems the loading of INU-NH₂ within INU-NH₂&E-NH₂@NKCOF-141, is different than for INU-NH₂@NKCOF-141, it may be relevant to include the loading of INU in INU-NH₂&E-NH₂@NKCOF-141 in the manuscript.

Response: We thank the reviewer for the comment. Per the suggestions, we have added the loading of INU-NH₂ (1.34 g g⁻¹) in INU-NH₂&E-NH₂@NKCOF-141 in the manuscript. The corresponding changes have been highlighted in yellow in the main text.

Reviewers' Comments:

Reviewer #1:

Remarks to the Author:

The authors have addressed my concerns. I recommend acceptance of the manuscript.